# Secondary Traumatic Stress in Portuguese Social Workers

Maria Irene Carvalho [1],[*] , Sofia Mendes Cunha [2], Helena Teles [1] and Carla Ribeirinho [1]

1 Instituto Superior de Ciências Sociais e Políticas, Centro de Administração e Políticas Públicas, Unidade de Coordenação de Serviço Social e Política Social, Universidade de Lisboa, 1300-663 Lisbon, Portugal
2 Centro Hospitalar de Lisboa Ocidental EPE Lisboa, 1169-050 Lisbon, Portugal
* Correspondence: mcarvalho@iscsp.ulisboa.pt

**Abstract:** This article analyses the level of secondary traumatic stress in Portuguese social workers. Social workers practice their profession in conditions that generate stress and trauma on a daily basis; this is triggered by clients' anxiety, threats to the integrity of the professionals, and situations that have a negative impact on professional practice. This study was conducted in organisations registered under the Social Charter, which coordinates institutions of social policies operationalised in Portugal. A Secondary Traumatic Stress (STS) index was used to obtain data regarding practitioners' practice for the characterisation and identification of situations of trauma experienced by clients, and their impact on social workers. Questionnaires were sent to the organisations via email, and in return 872 were answered by social workers. The results reveal a score of 2.65 on the Secondary Traumatic Stress Scale, and this appears more apparent in the Arousal dimension, followed by Avoidance and lastly Intrusion. Furthermore, it is evident that the conditions of professional practice and the sector in which professionals work generate STS because they significantly expose social workers to the traumas of the most vulnerable clients. The professionals recognise these incidences, which feasibly denotes that there is a possibility to avoid and act against STS by stimulating professionals' internal resilience and psychological capital, increasing professionals' access to professional supervision and improving working conditions in the organisations where they practice.

**Keywords:** secondary traumatic stress; social workers; trauma; Portugal

## 1. Introduction

Social work is a profession centred on human relations; therefore, social workers "listen" and provide psychosocial support to people in general and to clients on an ongoing basis. Beyond that, it is a profession of social intervention and is politically natured, given that it is exercised in the general society and within the welfare systems of states. It is positioned within the framework of public and social policies/institutions, bearing social obligations, and operating with relative autonomy and under the influence of robust theoretical, methodological and ethical underpinnings. The professional action undertaken within or outside the scope of these programmes focuses on the fight against inequality and oppression in society and the promotion of rights and social justice for persons, groups and communities who are vulnerable to social ills, such as poverty, unemployment, loneliness, isolation, physical and psychological aggression, sexual violence, human trafficking, diseases, loss/mourning, disasters and natural catastrophes, among others.

Social workers assume a key role in caring for such vulnerable populations and maintaining their wellbeing (Singer et al. 2020). In doing so, they observe and encounter traumatic events from their clients' "reports". For clients, communicating these upsetting events helps them in the process of healing and overcoming trauma, and yet this can affect the social workers who listen to and accompany them. Due to the constant exposure to these events experienced and reported by clients, social workers subsequently constitute a professional class at risk of developing secondary traumatic stress (STS).

STS is associated with behaviours and emotions that professionals experience arising from the helping process (Jacobs et al. 2019; Rauvola et al. 2019). STS is caused by the perceptual effects that professionals acquire from both the traumatic event itself as experienced by another person/client of the services, and the accompaniment and intervention made by these professionals in trying to overcome the event (Figley 1995, 1998, 2012; Moreno-Jiménez et al. 2004).

Secondary traumatic stress was defined by Figley (1999) as "the natural consequent behaviours and emotions resulting from knowledge about a traumatising event experienced by a significant other. It is the stress resulting from helping or wanting to help a traumatised or suffering person" (p. 10). It is a type of situation associated with stress and trauma (Figley 2012).

Stress is "an unpleasant state of strain, tension, or being taxed. In stress, physical, psychological, and social life-enhancing processes are challenged but not permanently dislodged. The source of stress is called a stressor, and the person is said to be stressed" (Figley 2012, p. 676). Stress can be resolved through a process of inner rebalancing and can be ameliorated. One kind of stress stems from actively listening to the events and during the process of helping or trying to help a traumatised or suffering person (Kleber and Figley 1995). This conceptualised stress originates from almost an impossible distance between those who suffer harm (clients suffering from primary traumatic stress) and those who care for them (professionals who are the custodians of the information of the traumatising event).

In the DSM-V TR (American Psychiatric Association 2022) the trauma is linked with "stressor-related ( . . . ) disorders in which exposure to a traumatic or stressful event is listed explicitly as a diagnostic criterion. These include reactive attachment disorder, disinhibited social engagement disorder, posttraumatic stress disorder (PTSD), acute stress disorder, adjustment disorders, and prolonged grief disorder" (p. 295). The symptoms of STS are similar as those of posttraumatic stress disorder (*PTSD*) (Diagnostic and Statistical Manual of Psychiatric Disorders DSM-IV; DSM-V 2022).

Social work in traumatic contexts can lead to unhealthy habits, impact the physical and mental health of professionals, negatively affect professional decision making, and consequently may rupture and/or lead to abandonment of the profession due to the inability to cope with the traumas and the stress surrounding these events (Armes et al. 2020; Bride 2007; Choi 2011, 2017; Harker et al. 2016). The psychological effects of STS may be roughly equal to the effects of direct trauma exposure (Bride et al. 2004); that is to say, the symptoms of secondary trauma (felt by professionals) may be similar to the symptoms seen in individuals who have been directly exposed to trauma. The psychological effects of direct exposure to extreme stressful and traumatic situations have been documented by several authors such as Carvalho (2011), Bride et al. (2004, 2007), and Figley (1999, 2012).

The analysis of the frequency of these symptoms among professionals enables to ascertain the level of secondary traumatic stress. Bride et al. (2004) categorises these symptoms into three: (1) symptoms associated with intrusive thoughts; (2) symptoms associated with avoidance attitudes; and (3) symptoms related to high levels of restlessness/nervousness. As highlighted before, the indirect exposure to trauma engenders a risk of emotional, cognitive and behavioural changes in the professional him/herself, thus STS is conceptualised as an occupational hazard for professionals dealing with people with primary trauma (Bride et al. 2007, p. 155). This implies that STS may endanger the worker's physical or psychosocial wellbeing and generally negatively impact organisational activities.

From the literature review, several studies unanimously contend that STS is frequent in human service professionals (de Castro et al. 2018; Harker et al. 2016; Schuler et al. 2016), and other studies emphatically point out social work professionals (Lev et al. 2022; Masson and Moodley 2020; Singer et al. 2020; Vîrgă et al. 2020; Owens-King 2019; Quinn et al. 2019; Choi 2017; Wagaman et al. 2015; Bride 2007; Simon et al. 2005; Bride et al. 2004).

Harker et al.'s (2016) study that focused on human service professionals in general disclosed that "higher levels of resilience were a significant predictor of lower levels of

psychological distress, burnout and secondary traumatic stress. Additionally, higher levels of mindfulness were a significant predictor of lower levels of psychological distress and burnout" (p. 631). The same study infers that STS is one of the reasons why many of these professionals leave the professional field. Characteristically, professionals with less resilience to traumatic situations and those working in the private sector tend to be more vulnerable to STS (Schuler et al. 2016).

Social workers work in human services, and the cases they handle influence the profession both positively and negatively. In Bride's (2007) research in the USA, social workers faced a high risk of developing STS, as many experienced some post-traumatic stress symptoms, with 15.2% suffering severely. This percentage was much higher than the 7.8% representative of the persons in the entire population who experience post-traumatic stress during their lifetime (Bride 2007). The same study further revealed that 40% of social workers thought repeatedly and unintentionally about the traumas experienced by their clients, 28% reported facing difficulties in concentrating and 26% felt emotionally numb (Bride 2007). It concluded that this type of stress has negative impacts on the quality of care provided by social workers, because it increases the risk of professionals making inappropriate professional judgements, misdiagnoses of social issues, and inappropriate individual interventions and treatment plans, and such mistakes reinforce the desertion of the profession.

Other studies conducted by Masson and Moodley (2020), Singer et al. (2020), Vîrgă et al. (2020), Owens-King (2019), Choi (2017), Wagaman et al. (2015) and Simon et al. (2005) broaden the context of STS among social workers. It is stated that STS is severe when professionals intervene explicitly in situations of intense risk and trauma, such as in disaster situations (Naturale 2007; Hodgkinson and Shepard 1994), oncology (Simon et al. 2005), various forms of violence and military services (Masson and Moodley 2020; Owens-King 2019; Choi 2011, 2017), and in child and youth protection (Armes et al. 2020; Badger et al. 2008).

Hodgkinson and Shepard (1994) revealed that 60% of social workers practising in disaster contexts had psychological disorders after one year. In sexual violence cases, it is stated that 57.1% experience intrusion as a result of the helping process with traumatised clients. Trauma from sexual abuse situations may interfere with the professional's own sexual experiences, which may cause deep distress and suffering to them (Choi 2011, 2017). Among clinical social workers working with children at risk and in danger, the main causes of secondary traumatic stress were identified as the suffering, commitment, and emotional involvement of professionals, in the defence of the best interests of children and young people (Armes et al. 2020). Similarly, this phenomenon is stated by Borjanić Bolić (2019).

Emotional involvement is an important characteristic of this profession, but this value (emotional involvement/empathy) can be ambivalent in these cases and cannot itself represent an STS predictor (Rauvola et al. 2019). The idea of a relationship between empathy and STS, and other traumas, has been questioned, especially by studies of organizational psychology that deal with issues of professional vocation. Rauvola et al. (2019) state that "empathy-based stress is relevant to any individual in an occupational role exposed to STS" (p. 299), as it can work as a positive factor to deal with the traumas experienced by clients.

In a recent study in Israel, Lev et al. (2022) established that factors such as lower levels of resilience and social support, higher frequency of ethical conflicts, and an increased exposure to violence by clients, are directly linked to higher levels of secondary traumatic stress in social workers.

To prevent or ameliorate STS, it is important to act on several fronts. For frontline professionals, it is important to recognise the signs and symptoms, that is, intrusive thoughts, avoidance attitudes and restlessness/nervousness when faced with responsibilities to intervene in traumatic cases (Bride et al. 2007; Figley 1999). Therefore, it should be known that interventions should be tailored towards combating rumination, this being the psychological stress response that involves a repetitive and persistent focus on symptoms associated with stress and various negative feelings (Menezes et al. 2017). This response is often linked to high levels of anxiety, depression and stress (Ferreira 2020). Therefore, support

should be provided to professionals through education or the supervisory processes to raise awareness concerning situations that often trigger STS, and to develop strategies to combat the occurrence of this type of trauma (Samson et al. 2022). For example, Sprang and Garcia (2022) provide evidence that "increased use of trauma-informed care practices can positively impact the STS levels of professionals ( . . . ), though special attention should be paid to those with high levels of intrusion or arousal" (p. 1095). Furthermore, Samson et al. (2022) points out that interventions focused on combating rumination and increasing social workers' awareness of their personal susceptibility could be effective in preventing STS.

STS can be mitigated, especially through deploying professional supervision strategies that stimulate internal personal changes, accelerate professionals' resilience (Owens-King 2019), increase self-centred psychological capital, efficacy, resilience, hope, and optimism (Lev et al. 2022; Vîrgă et al. 2020; Choi 2017; Harker et al. 2016; Kleber and Figley 1995), and encourage the development of empathic processes (Wagaman et al. 2015).

It is also important to improve working conditions in employing organisations by lessening caseloads (Quinn et al. 2019; Choi 2017; Schuler et al. 2016) and providing effective support supervision in intervention processes through monitoring workplace practices, training and strengthening social workers' resilience, and handling ethical conflicts effectively (Lev et al. 2022).

Employers and professionals should be concerned about the quality of services that clients access because their ineffectiveness in responding to needs can generate verbal and physical violence from clients towards social workers, which can in turn trigger secondary traumatic stress. It is therefore crucial to promote an ethical and positive climate in social work organisations and to establish adequate support processes for practicing social workers.

## 2. Methodology

In this article, we intended to analyse the level of STS in social workers in Portugal, and as well contribute to the discussions on the subject matter at an international level. Social workers in Portugal practice their profession in public and private non-profit social welfare institutions, largely in the sectors of social security, health, and education (Carvalho 2020). They are majorly involved in social action and the enhancement of people's wellbeing through their efforts to address common social concerns, that is, poverty and exclusion, loneliness and isolation, mental illness, unemployment, children and young people at risk and danger, chronic diseases, addiction/addictive behaviour, domestic violence, disability, school failure and dropout, discrimination, and homelessness and reclusion, among others (Carvalho 2020). Direct intervention stands out as their main approach in diagnostic assessments, social monitoring processes, enabling access to social rights in social policy programmes, and the entire approach when dealing with the population (Carvalho 2020). The indirect interventions focus on the management and coordination of responses, though falls considerably under the influence of managerialism (Carvalho 2020).

We used the Bride et al. (2004) Secondary Traumatic Stress Scale, a reliable and valid instrument that is specifically designed to measure the negative effects of secondary exposure to traumatic events. This instrument is also easy to apply and interpret. The scale is a 17-item instrument that assesses the frequency of intrusion, avoidance and arousal symptoms associated with secondary traumatic stress (Bride et al. 2004). The items are answered on a five-point Likert scale (1–5) ranging from 'never/' to 'very often' and correspond to the respondents' feelings over the past seven days.

Each item in this index corresponds to one of the 17 symptoms of STS: Intrusion (items 2, 3, 6, 10, 13), Avoidance (items 1, 5, 7, 9, 12, 14, 17) and Arousal (items 4, 8, 11, 15, 16). The score for the STS and for each subscale is obtained by summing the items assigned to each subscale from 17 to 85, with higher scores indicating higher level of STS.

The STS scale differs from the many available STS analysis measures, in that the wording of the instructions and the stems of the stressor-specific items (items 2, 3, 6, 10, 12, 13, 14, 17) were designed such that the traumatic stressor is uniquely represented by

exposure to clients. Other items are not stressor specific (items 1, 4, 5, 7, 8, 9, 11, 15, 16), but are characteristic of the negative effects of secondary traumatic stress.

Beside the STS data, the sociodemographic information and the fields of intervention, work sector, direct and indirect professional actions, stability in their workplace, income, number of cases handled, perception on traumatic event, and impacting traumas are also revealed.

A questionnaire was formulated and sent to the email addresses of social workers practising in institutions incorporated in the country's social network—a network that subsumes all institutions with social interventions in Portugal. Around 18,000 emails were sent to these organisations and to their respective social work professionals. The questionnaire, available for two months (between June and July 2021) was directly accessed by social work professionals and was answered online, and 872 valid responses were garnered. The proportion of responses was 4.8%.

The data were compiled in an Excel database, coded and treated with the SPSS program (version 27). Using this program, descriptive and psychometric analyses were performed and means, medians and standard deviations were calculated to generate the internal consistency of the scale. For reliability, Cronbach's alpha, standard deviation and correlations were also calculated, and lastly Pearson's Bivariate correlation for parametric samples was chosen as ideal, and multivariate analysis was also used.

With regard to ethics, this study was not submitted for approval by the ethics committee, given that it is not a requirement by the university research centre. Nevertheless, the respondents gave their informed consent at the beginning of answering the questions in the questionnaire. To this end, respondents marked the item "yes" to consent to their participation in the study, agreeing with the purpose of the research and expressing their willingness and freedom to participate in this research. It is also significant to mention that no conflict of interest was found between the parties involved, that is, between the researchers and the respondents.

## 3. Sample Profile

The participants in the study are predominantly female (95.5%), aged from 22 to 67 years, with the mean age being 42.55 years, which corresponds with other studies that map the profession in Portugal (Carvalho 2020). The most common age groups were between 40 to 49 years (36.9%), and 30 to 39 years (28.9%), and the vast majority are married (48.5%). For all social workers, the basic education is a degree in Social Work, as this is the prerequisite that qualifies them to practice the profession. Of these, around 40% refer to having post-graduate training, 22.1% in the form of specialised training and 16.3% of a master's degree. These professionals exercise their profession in the institutions of social action (54.8%), health (21.7%) and social security (9.7%). They possess professional experience ranging between 1 month and 34 years, with a calculable average of 7.7 years, though the most evident groups are between 11–15 years (20.4%), 16–20 years (16.5%) and 21–25 years (16.2%).

49.0% of the professionals provide services in the private sector and 39.7% in the public sector. Professionals working directly with clients form 64.1%, followed by indirect workers such as directors (23.5%) and service coordinators (11.7%). Participants had spent between 1 month and 34 years in their current workplace with 6.35 years being the average time spent, and those that spent up to 5 years stood out at 33.9%, followed by 11 to 15 years at 18.2%. Many professionals either possessed fixed term employment contracts (61.1%) or open-ended employment contracts (18.6%), which reveals a relative professional stability in their workplace. The most common income ranged between 901 to 1200 euros (33.6%), followed by between 1201 to 1500 euros (27.4%). In relation to professionals' hours of work per day, 49.4% worked for 7 h while 50.6% worked for 8 h (it should be noted that 61.2% of the professionals reported working five extra hours per week.

Moreover, 44.7% of the professionals consider themselves as having appropriate responsibilities for the functions they perform, whilst 49% believe they have more respon-

sibilities than they should be having. Variables relating to the number of cases handled vary: 0.0 for service coordinators and 268 for professionals in direct intervention, with the average number of cases being 27.2 for the latter. These averages were grouped together whereupon those with less than 20 cases were 66.7%, and those with between 20 and 49 cases were 13.5%. The number of people involved in these cases are between 1 and 720, with an average of 75.77 people per professional. These data were further grouped whereupon findings showed that 31.4% have more than 80 people, 23.5% have less than 20 people, 22.7% have between 20 and 49 people and 18% have between 50 and 79 people, thus revealing an excessive ratio of clients per professional.

Professionals reported some traumatic incidents during their professional experience. The traumatic events were a result of prolonged illness of the clients or their relatives (50.5%), the unexpected or sudden death of the client or relative (37.2%), adult victims of violence (36%), and child neglect (31.8%). Considering the information and perceptions we have obtained, we subsequently present and analyse the results of the scale to understand the level of STS among Portuguese social workers.

## 4. Results STS Index

The descriptive analyses revealed that the STS scores were between 18 and 85. According to Bride (2007), a score is classified as moderate (17–39 points); high (40–62 points); or severe (63–85 points). In this research the average was 44,9 which indicates high levels of stress, but not very high. Table 1 shows that 23.39% and 23.50% have never or rarely experienced STS, 26.40% occasionally, and 18.40% and 8.24% often and very often (as shown in Table 1).

If we take into account the mean of each item, in Table 1 we can notice that the highest weighting is item (10) "I thought about my work with clients/users even without intending to" (3.34); then item (11) "I had difficulties concentrating" (3.02); item (4) "I had difficulties sleeping" (2.92); item (15) "I felt easily irritable" (2.88), and item (17) "I noticed gaps in my memory while working with clients/users" (2.83). Item (10) and (17) relate to the exposure of clients to trauma and they have the most impact on secondary traumatic stress. On the other hand, items (11) and (15) are related to the negative effects of STS, and these have the severest effects.

The following items had the lowest weighting: (13) "I had disturbing dreams about my work with clients/users" (2.19); (6) "Being reminded about my work with clients/users made me feel disturbed" (2.27); and (16) "I felt that something bad would happen" (2.25). Thus, items (13) and (6) relate to exposure to clients' trauma and these have the least impact on STS. Item (16) is related to the negative effects, and it has the least effect.

The factorial analysis shows the consistency and the average of the Secondary Traumatic Stress Index—Table 2. The Cronbach's Alpha is 0.950, which reveals a high internal consistency of STS since the "acceptable" value for social sciences is 0.70. The overall mean is 2.65, which indicates that STS is "trending" among these professionals. For these professionals, the STS is more expressive in the Arousal dimension (2.91 average), followed by Avoidance (2.63 average), and finally Intrusion (2.56 average), as demonstrated in Table 2 below.

The correlation between STS (Avoidance, Arousal and Intrusion) and other dimensions such as age group, area of intervention, field work, number of social processes, intervening in crisis situations, and the type of trauma revealed by these professionals indicate that these variables have positive low (r—0.10–0.29) and moderate (r—0.30–0.49) correlations (Cohen 1988). Table 3 shows only few have a high correlation (r—0.50–1).

**Table 1.** Statistics of STS.

| | Never (1) | Rarely (2) | Occasionally (3) | Often (4) | Very Often (5) | Factorial Analyses Average (1–5) |
|---|---|---|---|---|---|---|
| (1) I felt emotionally numb | 11.4% | 23.7% | 42.4% | 17.8% | 4.7% | 2.81 |
| (2) I had headaches when I thought about the work I do with clients/users | 23.3% | 23.5% | 26.8% | 17.7% | 8.7% | 2.65 |
| (3) I felt as if I was feeling the trauma experienced by clients/users | 28.2% | 30.0% | 24.7% | 13.% | 4.1% | 2.35 |
| (4) I had trouble sleeping | 18.7% | 21.4% | 22.9% | 22.7% | 14.2% | 2.92 |
| (5) I felt discouraged about the future | 17.5% | 22.0% | 30.8% | 21.2% | 8.4% | 2.81 |
| (6) Being reminded about my work with clients made me feel upset | 31.4% | 29.8% | 23.1% | 11.9% | 3.8% | 2.27 |
| (7) I felt little interest when being surrounded by other people | 30.6% | 25.1% | 26.6% | 11.7% | 6.0% | 2.37 |
| (8) Felt nervous and/or startled | 19.6% | 25.3% | 27.2% | 19.5% | 8.4% | 2.72 |
| (9) Been less active than usual | 21.4% | 22.8% | 32.6% | 17.9% | 5.3% | 2.63 |
| (10) I thought about my work with clients/users, even without intending to | 9.9% | 14.0% | 27.4% | 30.2% | 18.6% | 3.34 |
| (11) Had trouble concentrating | 12.5% | 19.4% | 32.7% | 24.5% | 10.9% | 3.20 |
| (12) Avoided people, places or things that reminded me of my work with clients/users | 32.% | 22.9% | 19.3% | 15.9% | 9.9% | 2.49 |
| (13) I had disturbing dreams about my work with clients/users | 38.6% | 26.1% | 18.6% | 10.8% | 5.8% | 2.19 |
| (14) I felt like avoiding working with certain clients/clients | 30.7% | 25.0% | 20.2% | 16.7% | 7.3% | 2.45 |
| (15) I was easily irritated | 16.4% | 22.2% | 28.0% | 23.4% | 10.0% | 2.88 |
| (16) Felt that something bad was going to happen | 35.7% | 25.2% | 20.9% | 14.4% | 3.8% | 2.25 |
| (17) I noticed gaps in my memory while working with clients/users | 19.8% | 21.7% | 24.5% | 23.7% | 10.2% | 2.83 |
| Total | 23.39% | 23.0% | 26.4% | 18.4% | 8.24% | 2.65 |

**Table 2.** Mean and standard deviation.

| | Mean | Standard Deviation |
|---|---|---|
| Total STS | 2.65 | 0.90322 |
| Intrusion—items 2, 3, 6, 10, 13 | 2.56 | 0.95163 |
| Avoidance—items 1, 5, 7, 9, 12, 14, 17 | 2.63 | 0.91068 |
| Arousal—items 4, 8, 11, 15, 16 | 2.91 | 1.04539 |
| Post Traumatic Stressor Items | 2.57 | 0.94080 |
| Items of the negative effects of post-traumatic stress | 2.71 | 0.92899 |

**Table 3.** Correlations.

| | | Age Group | Social Intervention Field | Sector of Intervention | Direct/Indirect Professional Responsibilities | Average, Number of Processes Accompany Daily | T1—Trauma Adults' Victims of Physical Violence | T2—Trauma Related to Prolonged Illness of a Family Member | T3—Trauma and Victims of Neglect | T4—Trauma Related to Unexpected or Sudden Death of a Family Member | Intrusion | Avoidance | Arousal | Total—STS |
|---|---|---|---|---|---|---|---|---|---|---|---|---|---|---|
| Age group | Pearson's correlation | 1 | | | | | | | | | | | | |
| | Sig. (2 ends) | | | | | | | | | | | | | |
| Social intervention field | Pearson's correlation | 0.209 ** | 1 | | | | | | | | | | | |
| | Sig. (2 ends) | 0.000 | | | | | | | | | | | | |
| Sector of intervention | Pearson's correlation | −0.317 ** | −0.271 ** | 1 | | | | | | | | | | |
| | Sig. (2 ends) | 0.000 | 0.000 | | | | | | | | | | | |
| Direct/indirect professional responsibilities | Pearson's correlation | 0.060 | −0.213 ** | 0.310 ** | 1 | | | | | | | | | |
| | Sig. (2 ends) | 0.075 | 0.000 | 0.000 | | | | | | | | | | |
| Average number of processes accompanied daily | Pearson's correlation | 0.021 | −0.001 | 0.082 * | 0.169 ** | 1 | | | | | | | | |
| | Sig. (2 ends) | 0.541 | 0.986 | 0.015 | 0.000 | | | | | | | | | |
| T1—Trauma adults victims of physical violence | Pearson's correlation | 0.047 | 0.000 | 0.058 | 0.177 ** | 0.064 | 1 | | | | | | | |
| | Sig. (2 ends) | 0.169 | 0.991 | 0.084 | 0.000 | 0.059 | | | | | | | | |
| T2—Trauma related to prolonged illness of a family member | Pearson's correlation | 0.005 | 0.012 | −0.002 | −0.069 * | 0.046 | 0.132 ** | 1 | | | | | | |
| | Sig. (2 ends) | 0.879 | 0.728 | 0.948 | 0.041 | 0.179 | 0.000 | | | | | | | |
| T3—Trauma and victims of neglect | Pearson's correlation | −0.062 | −0.086 * | 0.126 ** | 0.196 ** | 0.044 | 0.207 ** | −0.038 | 1 | | | | | |
| | Sig. (2 ends) | 0.067 | 0.011 | 0.000 | 0.000 | 0.195 | 0.000 | 0.259 | | | | | | |
| T4—Trauma related to unexpected or sudden death of a family member | Pearson's correlation | −0.002 | 0.049 | −0.010 | −0.083 * | 0.031 | 0.105** | 0.539 ** | −0.015 | 1 | | | | |
| | Sig. (2 ends) | 0.947 | 0.150 | 0.767 | 0.014 | 0.360 | 0.002 | 0.000 | 0.660 | | | | | |

Table 3. *Cont*.

| | | Age Group | Social Intervention Field | Sector of Intervention | Direct/Indirect Professional Responsibilities | Average, Number of Processes Accompany Daily | T1—Trauma Adults' Victims of Physical Violence | T2—Trauma Related to Prolonged Illness of a Family Member | T3—Trauma and Victims of Neglect | T4—Trauma Related to Unexpected or Sudden Death of a Family Member | Intrusion | Avoidance | Arousal | Total—STS |
|---|---|---|---|---|---|---|---|---|---|---|---|---|---|---|
| **Correlations** | | | | | | | | | | | | | | |
| Intrusion | Pearson's correlation | 0.005 | −0.131 ** | 0.100 ** | 0.135 ** | 0.049 | −0.027 | −0.066 | −0.053 | −0.071 * | 1 | | | |
| | Sig. (2 ends) | 0.885 | 0.000 | 0.003 | 0.000 | 0.149 | 0.422 | 0.051 | 0.117 | 0.036 | | | | |
| Avoidance | Pearson's correlation | 0.022 | −0.116 ** | 0.094 ** | 0.065 | 0.005 | −0.047 | −0.087 * | −0.049 | −0.099 ** | 0.822 ** | 1 | | |
| | Sig. (2 ends) | 0.512 | 0.001 | 0.006 | 0.053 | 0.891 | 0.165 | 0.010 | 0.149 | 0.004 | 0.000 | | | |
| Arousal | Pearson's correlation | −0.029 | −0.138 ** | 0.114 ** | 0.129 ** | 0.032 | −0.040 | −0.051 | −0.022 | −0.084 * | 0.816 ** | 0.859 ** | 1 | |
| | Sig. (2 ends) | 0.399 | 0.000 | 0.001 | 0.000 | 0.344 | 0.232 | 0.130 | 0.510 | 0.013 | 0.000 | 0.000 | | |
| Total—STS | Pearson's correlation | 0.005 | −0.138 ** | 0.107 ** | 0.115 ** | 0.029 | −0.041 | −0.075 * | −0.046 | −0.090** | 0.926 ** | 0.959 ** | 0.933 ** | 1 |
| | Sig. (2 ends) | 0.872 | 0.000 | 0.002 | 0.001 | 0.386 | 0.223 | 0.027 | 0.177 | 0.008 | 0.000 | 0.000 | 0.000 | |

** The correlation is significant at level 0.01 (2 extremities). * The correlation is significant at the 0.05 level (2 extremities).

Some variables show a moderate correlation: T4—Trauma related to unexpected or sudden death of a family member; T2—Trauma related to prolonged illness of a family member (0.539); and work sector with direct and indirect intervention (0.310). These first two types of trauma are related and the sector of work and social work position are also important. These variables explain social workers STS.

To understand these results, we made a multivariate analysis of principal components. This analyses reduces the number of variables to explain the phenomenon. When we performed this reduction, five items emerged from the scale: 1, 2, 3, 4, and 5, with $p < 0.10$. Taking this information into account, we profiled social workers from never/rarely to high STS, shown in Table 4 below. In this analysis, variables emerge, such as the age group, area of interventions, and sector, as relevant to the understanding of STS in the profession (the number of cases are not relevant in the two categorized profiles). The number of process and intervention type are not relevant in the two categorized profiles.

**Table 4.** Profile 1 and 2.

| Categories | Indicators | Profile 1—Positive Taking STS into Account (Often) | Profile 2—Negative Taking STS into Account (Never/Rarely) |
|---|---|---|---|
| Age group | 20–29 years | | −1.524 |
| | 30–39 years | | −0.946 |
| | 50–59 years | 1.261 | |
| Area of intervention | Social action | | −0.880 |
| | Health | 1.056 | |
| | Social security | 1.234 | |
| Sector | Private | 0.768 | |
| | Private/public | 0.987 | |
| | Public | | −1.230 |

Profile 1 comprises social workers who responded highly to STS-Item 1 to 5: the age group 50–59 years, work in health and social security fields, and work in public and private sectors. Most of the services are provided by non-profit organisations under public–private partnership, as is the case in many areas of health and social security. Profile 2 comprises of professionals who responded never/rarely to STS-item 1 to 5: the age group of 20–29 and 30–39, area of intervention of social action and work in the public sector.

## 5. Discussion

In this study, social workers were predominantly female, and perform their work both in public/state and private/non-state settings within the welfare system in Portugal. They are professionally secure in their jobs but earn a low average income compared to other professions such as medicine and engineering. They mostly work directly with individuals and groups and deal with a high number of cases, which translates into relatively high client proportions. This overload is evident in the number of clients that these professionals attend to and the extra responsibilities assigned to them, since about half of these professionals disclose having more responsibilities than stipulated.

Many social workers who participated in this study believe they deal with traumatic situations on a daily basis. The level of STS is tendentially moderate for these professionals. This stems from handling or listening to lived experiences of clients that involve death, violence and children/youth victims of childhood/youth neglect. The above finding is consistent with investigations conducted by Lev et al. (2022), Masson and Moodley (2020), Singer et al. (2020), Vîrgă et al. (2020), Owens-King (2019), Quinn et al. (2019), Choi (2017), Wagaman et al. (2015), Bride (2007), Simon et al. (2005) and Bride et al. (2004).

There are two aspects that stand out in this study as triggers of STS: the field in which they practice and the type of trauma that clients experience. However, the former has more impact on the severity of STS. The most impacting daily traumas occur in professionals who work directly with clients and family members and are faced with pain arising

from the death of a client or family member or traumatic situations related to children at risk and danger. These specific situations of extreme vulnerability generating secondary traumatic stress in social workers are also captured in the literature (Armes et al. 2020; Badger et al. 2008).

These professionals repeatedly and unintentionally think about the traumas experienced by their clients and reveal having difficulty sleeping soundly, especially when they increasingly face challenges that generate work stress, such as handling complex problems of clients, having heavy workloads, working in unfavourable physical environments, and encountering emotionally stressful environments.

The percentage of social workers experiencing severe symptoms of STS is significant as captured in Table 2, that is 2.65 (1–5). Social workers are systematically confronted with traumatic accounts during their professional practice and the impact of these situations on their mental health has been less explored. Social workers reveal facing more difficulties in dealing with Arousal (2.91), that is, thinking about clients' problems, having difficulties with sleep and becoming more irritable. These STS factors are mostly influenced by the sector and field of practice rather than the number of social processes they deal with on a daily basis. For this reason, Choi (2017) argued that, in such circumstances, organisational policies are essential in alleviating STS among the infirm social work professionals.

The exploratory study also reveals more interesting data on how the STS is not the same for all professionals. For example, principal component analysis demonstrates that not everyone has the same weight in the analysis. So, in addition to the main and general characteristics, there may be two profiles that stand out: one where the STS is moderate, and the other where the STS is frequent. Variables such as age group, area of intervention, and sector where they work influenced this analysis.

The findings of this research are not new and are validated by Schuler et al. (2016), who demonstrated that professionals working in the private sector are more prone to STS. The fact that the state is no longer a provider of direct services and gives this responsibility to private organizations puts these organizations and professionals at risk of STS. Social workers are the frontline workers in these organizations as they provide services to vulnerable groups.

Despite these structural difficulties, social workers have demonstrated their ability to adapt to crises and contributed to overcoming these difficult situations. Being aware of the risk and protective mechanisms against STS is an important condition for the wellbeing of professionals and the quality of interventions they develop for citizens at national and international level. However, as Holmes et al. (2021) asserts, it is essential that the organisations provide adequate resources and continuous support for the emotional wellbeing of their employees.

## 6. Conclusions

In this study, we intended to analyse the level of STS in social workers of Portugal, and also contribute to discussions on the subject at international level. We supported the notion that the consequences of traumatic events are not only limited to those who directly experience the events (clients) but may also affect the professionals in close contact with them, who may develop symptoms classified as secondary traumatic stress. Several studies have been carried out at international level highlighting how critical this subject is for the profession and generally for the quality of services provided by these professionals in a social welfare system.

The STS comes from various sources: the clients' traumas, the type of relationship established between the social worker and the client, the working conditions, and the field where the professional practices his/her profession, though the sector and the area of intervention are the most relevant. Professionals increasingly face challenges that contribute to work stress, that is, having heavy workloads, working in unfavourable physical environments, and encountering emotionally stressful environments. Additionally,

the state of the broad social and economic environment is another contributing factor that this study discerns.

In recent years, it is noticeable that social workers are dealing with immense workloads, handling more cases and enduring extensive bureaucracy, and this is amidst high pressure from their employers to achieve goals that are not always in line with social realities (Guadalupe et al. 2020). Indeed, it is not surprising that these professionals end up experiencing STS.

In order to cope with and mitigate STS, it is important to act on several fronts: increase psychological capital (Vîrgă et al. 2020), increase resilience and empathy towards clients' traumas (Wagaman et al. 2015), demand decent working conditions from organisations, and access professional supervision support (Choi 2017). Thus, spaces are needed in which professionals themselves receive support to identify the conditions that lead to stress early and to discourage psychological and emotional fatigue (Aristu 2008), but also where they can collectively find strategies to prevent and overcome STS (Ribeirinho 2019).

Such a complex subject points to future research paths that can be explored, namely: analysing how STS affects social workers' work in the medium- and long-term period in the private sector; analyse whether the professionals' quality of life and social support are directly linked to a higher vulnerability to develop STS symptomatology; and assess the relationship between dimensions of STS (trauma, antecedents, personality and consequences) and the characteristics of social workers' work in specific contexts. It is also significant to replicate this study with a sample of professionals who had participated in the supervision processes, in order to assess the extent to which professional support and supervision acts as a protective factor in STS.

This study on STS in social workers potentially addresses a subject which has limited or no similar studies in Portugal. This underlines its relevance especially in these complex times we are living in, and with the density of issues which these professionals are faced with. Despite the evident stresses and tensions observed in the professionals, social work is a rich and rewarding profession, offering an important and positive contribution to society and therefore deserves to be valued. Conclusively, this research may also be useful for future international and comparative studies on the topic.

**Author Contributions:** Conceptualization, M.I.C.; methodology, H.T.; formal analysis, S.M.C.; investigation, M.I.C.; writing—review and editing, C.R.; All authors have read and agreed to the published version of the manuscript.

**Funding:** This research received no external funding.

**Informed Consent Statement:** Informed consent was obtained from all subjects involved in the study.

**Conflicts of Interest:** The authors declare no conflict of interest.

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
