# Peer review of "Secondary Traumatic Stress in Portuguese Social Workers"

_socsci, doi:10.3390/socsci12040240_

Round 1

Reviewer 1 Report

This work addresses an important topic, as social workers are clearly at high risk for exposure to Secondary Traumatic Stress.  It is a strength of this manuscript that it highlights not only the individual risk factors, but also the importance of organizational characteristics that are needed to support this workforce. The following suggestions are offered within the spirit of constructive criticism: 

  1. The introduction relies too heavily on sources from one author that are over ten years old. Although. Other studies are listed, they are not explored in depth.  In particular, consider a more widely accepted definition of trauma that the one provided related to the loss of life-enhancing processes.
  2. Note that within the DSM-5, PTSD could be diagnoses from a secondary trauma exposure, so it technically could be considered a mental illness. Consider reviewing Sprang et al., 2022, published in the Journal of Interpersonal Violence, for more information on how STS could be interpreted diagnostically. 
  3. In the literature review, provide more detail about the findings of the studies listed related to the prevalence of STS in human services professionals. For example, which symptoms are most frequent? What is the prevalence of STS at various levels?  A few studies are cited in detail, but are there reviews or a general trend/consensus?
  4. I recommend a more comprehensive and organized presentation on the risk and resilience factors in previous studies for STS.
  5.  The authors state that STS can be caused by emotional involvement and commitment, which is not generally supported by research that suggests that compassion and engagement with clients is a career-sustaining rather than harmful force. Consider revising the sentence on page 3, line 132-133, or at least putting this finding in context.
  6. The sentence on page 4, line 159-162 seems to suggest that social workers are to blame for their own violent victimization because of their poor quality of services, which to me is highly problematic.  
  7. In describing the STSS measurement tool, the authors state that it corresponds to the DSM-5 on line 186 of page 4, which is incorrect for the 17-item version. 
  8. The response rate reported of 100% seems misleading, since the typical percentage rate reported would be the percent of responses out of the number of questionnaires sent.  I suggest reporting the response rate out of the 18,000 surveys sent.
  9. Clearly state research questions that will be examined, and which analyses will be used for which questions. 
  10. When calculating averages of STS symptoms, do the authors remove participants who have caseloads of zero or who do not report trauma exposure? I recommend removing those without exposure, as their symptoms could not be considered STS. 
  11. It is important to clearly distinguish STS through exposure to client’s trauma from primary trauma, as a result of violence in the workplace or in one’s personal life.  These should be clearly distinguished so that the results of STS symptoms can be connected to secondary trauma and not PTSD as a result of direct trauma exposure.  This distinction should be made in the analyses, as well as when describing the results and conclusions.
  12. In reporting the means of different scales or items, what guidance is being followed in interpreting these numbers?  Consider reporting the totals and the descriptive categories published by Bride, et al (e.g., severe, high, moderate).
  13. Clarify the research question and analytic procedures used to report the different associations with STS symptoms and types of trauma exposure. I would assume many professionals would be exposed to multiple types, so how were you grouping those who’ve experienced only one type as opposed to another? 
  14. I’m not sure what the importance of the correlation between measures and items is, or how this connects with overall research questions. 

Author Response

Dear review

First of all, we would like to thank you for your excellent review of the article, which allows us to improve not only the part but above all the way the data are presented and consequently reformulate, albeit briefly, the discussion of the results. The text was also revised in the English language. We hope that the changes respond to the request. Best Regards. 

Reviewer 2 Report

  1. At the  bottom of page 4 (at the end of the line #$205), what does this "100%" mean?
  2. In Results of STS Index (page 6, 3rd paragraph), it is repeating the content reported at the bottom of the page 5 and the top of page 6; in the next paragraph, it is repeating the content reported in the 2nd paragraph of page 6.
  3. On page 7, the second and third paragraphs, there are no reports on the analyses of these findings in any tables.
  4. At the bottom of page 7, the authors will replace "significant correlation" with "strong correlation". In the same paragraph, what is "p" standing for in its report “p=0.961”? Where does value 0.961 come from? It is not reported in the table 2.
  5. On page 8, the first paragraph, what is "p" standing for its reports of values in lines #343 and #344? Where do those p's values come from? They are not reported in the table 2.
  6. In Conclusion (page 9, 4th paragraph), there is no report of analyses and results that support correlations between indirect service and STS. In the next paragraph, there is no report of analyses results that involved correlations between workload (or work conditions) and STS. Authors’ argument in these two paragraphs has no support from the study’s results.
  7. Authors will need multivariate analyses in which STS is the outcome variable and other variables (e.g., gender, age, experience, work conditions, direct/indirect service, types of trauma, etc.) as explanatory variables. Without such analyses, the argument in Discussion appears to be weakly based.
  8. In the paragraph at the bottom of page 9 (line# 435 to 437), authors will provide some specific examples how to increase psychological capital and resilience.
  9. At the top of page 10 (line# 441), authors will provide some specific examples of such strategies.

Author Response

(The authors gave the same response as above.)

Round 2

Reviewer 2 Report

1.      On page 6, the first paragraph, authors will replace all the commas with decimal points in the description of numerical values. Throughout the rest of manuscript, there are many more such mistakes. Also, authors will replace all the commas with decimal points in all tables.

2.      Authors will correctly label the table for STS Statistics with "Table 1".

Author Response

Dear Review

 First of all, I would like to thank you for reviewing the article and accepting it for publication.

Taking into account the notes, I made some changes:

Replacement of the comma by the dot in the tables and respective text was performed, as well as the

Replace all the commas with decimal points in all tables.

Table 1 was also replaced by its name - Statistics STS "Table 1".

The text suggested by the editor was included in begining of  the conclusion:

“In this study, we intended to analyze the level of STS in social workers in Portugal, and also contribute to the discussions on the subject at international level. We supported the notion that the consequences of traumatic events are not only limited to those who directly experience the events (clients) but may also affect the professionals in close contact with them, who may develop symptoms classified as Secondary Traumatic Stress”

The main alterations are marked in yellow, but in the case of points and hundredths, these alterations could not be marked because there were too many.

Regarding English, I would like to inform you that the article was revised twice by a person who is native to English, who is also a social worker and is originally from Uganda. I think it makes no sense to be reviewing again.

Best Regards

MIC
